# Trends in Survival and Mortality of “Early” Metastatic Breast Cancer in Northern Italy Following the Introduction of Targeted Therapies

**DOI:** 10.3390/cancers18010108

**Published:** 2025-12-29

**Authors:** Francesco Marinelli, Maria Barbara Braghiroli, Isabella Bisceglia, Guglielmo Ferrari, Fortunato Morabito, Filippo Giovanardi, Carmine Pinto, Lucia Mangone

**Affiliations:** 1Epidemiology Unit, Azienda USL-IRCCS di Reggio Emilia, 42122 Reggio Emilia, Italy; francesco.marinelli@ausl.re.it (F.M.); mariabarbara.braghiroli@ausl.re.it (M.B.B.); isabella.bisceglia@ausl.re.it (I.B.); 2Associazione “Vittorio Lodini per la Ricerca in Chirurgia” di Reggio Emilia, 42122 Reggio Emilia, Italy; ferrariguglielmo3@gmail.com; 3AIL Sezione di Cosenza, 87100 Cosenza, Italy; fortunato.morabito@grade.it; 4Medical Oncology Unit, Azienda USL-IRCCS di Reggio Emilia, 42122 Reggio Emilia, Italy; filippo.giovanardi@ausl.re.it (F.G.); carmine.pinto@ausl.re.it (C.P.)

**Keywords:** metastatic breast cancer, incidence and survival trends, early detection, precision oncology, real-world evidence

## Abstract

Breast cancer is the most common cancer among women, and its outcome largely depends on how early it is detected and how effectively it is treated. This study examined how the number of women diagnosed with metastatic breast cancer, and their chances of survival, have changed over the past two decades in the province of Reggio Emilia, Northern Italy. The region benefits from organized mammography screening and widespread access to modern, targeted treatments. The results show that fewer women are now diagnosed with “early” metastatic disease and that survival after diagnosis has improved. These findings demonstrate how the combination of early detection and advances in precision medicine can significantly improve outcomes for women with breast cancer, providing real-world evidence that progress in both screening and therapy is saving lives in the community.

## 1. Introduction

Breast cancer remains the most frequently diagnosed malignancy and the leading cause of cancer-related death among women worldwide. According to the most recent estimates from the World Health Organization (WHO) [1] and the Global Cancer Observatory (GLOBOCAN 2020) project [2], breast cancer accounts for approximately 24% of all new cancer diagnoses and 15% of cancer-related deaths in women.

Breast cancer incidence shows marked geographic heterogeneity. In high-income countries such as the United States, Canada, and much of Western Europe, incidence rates are stable or slightly decreasing, largely due to organized mammography screening programs, early diagnosis, and therapeutic advances [3,4].

Conversely, in middle- and low-income countries, incidence continues to rise, often in association with Westernized lifestyles (reduced physical activity, delayed childbearing, decreased breastfeeding, hypercaloric diets) and the lack of organized screening programs [5,6].

Despite remarkable therapeutic advances that have significantly improved 5-year overall survival—now exceeding 85% in industrialized countries—breast cancer continues to pose a major public health challenge [7]. In many settings, diagnosis still occurs at advanced stages, resulting in poorer prognosis and higher mortality [8].

Differences in incidence and mortality reflect not only biological and genetic factors but also social determinants and disparities in access to care. Consequently, current research and health policy efforts are increasingly focused on integrated strategies, encompassing primary prevention, early detection, and personalized treatment based on tumor molecular characteristics [9,10].

Metastatic breast cancers (MBCs) display substantial epidemiological variability across regions, depending on the geographic, socioeconomic, and healthcare context. This heterogeneity is strongly influenced by the availability and effectiveness of screening programs and secondary prevention initiatives [11,12]. In countries where organized mammography screening covers the target population (typically women aged 45 to 74), a significant reduction in diagnoses at the metastatic stage has been documented [13].

Early diagnosis enables the detection of lesions at an initial phase, allowing for less aggressive treatments and a more favorable prognosis [14,15].

Several studies have shown that systematic screening participation reduces breast cancer mortality by up to 30–40% [16,17]. Conversely, in settings where screening is absent, fragmented, or inaccessible—often due to infrastructural, economic, or sociocultural barriers—the proportion of patients presenting with metastatic disease is substantially higher [18]. In these settings, breast cancer is often diagnosed at advanced stages (III–IV), with negative implications for overall survival and quality of life [19]. Even within Italy, notable variability exists: the proportion of metastatic cases at diagnosis ranges from 6.5% to 7.1% overall, with regional differences—5.1% in the North, 7.4% in the Centre, and 7.8% in the South [20,21].

In recent years, several high-income countries have reported a decline in both incidence and mortality related to MBC. While the reduced incidence reflects the benefits of screening and early detection, a major contributing factor is the therapeutic revolution in the management of advanced disease. The introduction of targeted therapies, endocrine agents, CDK4/6 inhibitors, HER2-directed treatments, and, more recently, immunotherapies has markedly improved survival outcomes among patients with MBC [22,23]. In nations with broad and equitable access to these modern therapies—such as the United States, several European countries, and parts of East Asia—mortality rates for MBC have steadily declined over the past two decades [24,25,26,27].

This decline reflects not only improved access to systemic therapies but also a deeper understanding of tumor biology, routine molecular profiling, and the integration of multidisciplinary care. The sequential use of modern treatments—such as CDK4/6 inhibitors (palbociclib, ribociclib, abemaciclib) for hormone receptor-positive disease, trastuzumab, pertuzumab, T-DM1, and trastuzumab deruxtecan for HER2-positive subtypes, and immune checkpoint inhibitors for triple-negative breast cancers—has prolonged survival while maintaining quality of life [28,29,30]. Given the paucity of observational studies on the impact of new drugs in metastatic breast cancers, this study aims to fill a knowledge gap by attempting to provide some useful information for interpreting the efficacy of treatments.

This study aims to analyze changes in the incidence of MBC and associated mortality in a Northern Italian province characterized by high screening participation rates and broad access to modern therapeutic options.

## 2. Materials and Methods

This study includes 10,966 cases of breast cancer registered in the Cancer Registry (CR) of the Province of Reggio Emilia (Northern Italy) between 2000 and 2022. The registry covers a population of 532,000 inhabitants. The CR of Reggio Emilia, Italy, is characterized by high-quality data, with 98.8% of breast cancer microscopically confirmed and less than 0.1% identified from death certificates only [31]. The registry’s data is updated through 2022, and its operations are approved by the Provincial Ethics Committee of Reggio Emilia, Italy (Protocol no. 2014/0019740, 4 August 2014).

Data sources include anatomical pathology reports, hospital discharge records, and mortality data, supplemented by laboratory results, diagnostic imaging reports, and information from general practitioners. We reported the cause-specific mortality: since the Cancer Registry is closely linked to the Mortality Registry, it is easy for us to access the death records and verify the cause-specific mortality data.

Stage information is collected by registrars working at the CR. International registration rules state that registrars are not required to formulate diagnoses, but only to report the information they retrieve from medical records and letters to the doctor. This explains why our data always contain some Unknown data, not because the patients were not staged, but because we were unable to retrieve it. Tumor stage was classified according to the 8th edition of the TNM staging system [32]. 

An additional table has been inserted, which summarizes the list of drugs that were analyzed in the province of Reggio Emilia, Italy, in the period studied. Temporal trends were evaluated by calculating the annual percentage change (APC) in mortality using joinpoint regression analysis (Joinpoint Regression Program, Version 5.4.0.0), a method widely adopted in cancer epidemiology and applied by the SEER Cancer Statistics Review [33]. A segmented log-linear regression model was used. With this approach, the cancer rates are assumed to change at a constant percentage of the rate of the previous year. One advantage of characterizing trends this way is that it is a measure that is comparable across scales for both rare and common cancers. It is not always reasonable to expect that a single APC can accurately characterize the trend over an entire series of data. The joinpoint model uses statistical criteria to determine when and how often the APC changes. For cancer rates, it is fit using joined log-linear segments, so each segment can be characterized using an APC. Finding the joinpoint model that best fits the data allows us to determine how long the APC remained constant and when it changed. The maximum number of joinpoints allowed for these analyses was four [34]. Finally, the 1-, 3-, and 5-year net survival rates for cancers diagnosed between 2000 and 2022 were estimated. The study period was divided into three intervals: 2000–2007, 2008–2016, and 2017–2022. For the last interval, the calculation of 5-year net survival was restricted to cases diagnosed in 2017–2019 to ensure adequate follow-up. Net survival estimates overall survival adjusted for contributing causes of death and is defined as the ratio of the observed survival proportion in a cohort of cancer patients to the expected survival proportion in a comparable group of cancer-free individuals. Net survival was estimated using the Pohar Perme method, wherein net survival for a cohort is assessed by weighting with the inverse of the individual-specific expected survival probabilities. All statistical analyses were performed using STATA version 16.1 (StataCorp, College Station, TX, USA).

## 3. Results

Table 1 presents the number of breast cancer cases, including metastatic cases, recorded in the Province of Reggio Emilia from 2000 to 2022. Among 10,966 invasive malignant tumors, 511 cases (4.7%) were metastatic at diagnosis. The proportion of metastatic cases declined steadily from 6.4% (95% CI: 5.2; 7.6) in 2000–2003 to 3.8% (95% CI: 3.0; 4.6) in 2019–2022.

The 5.1% observed in 2022 could be linked to the fact that during the COVID-19 pandemic, all screening activities were temporarily suspended. In 2021 and 2022, screening progressively resumed, although some delays in new cancer diagnoses persisted. It will be important to assess whether the decline in metastatic tumors remains significant in 2023, thereby confirming the pre–COVID-19 trend

Table 2 summarizes deaths occurring within 1 and 2 years after diagnosis among patients with MBC. A total of 177 deaths (34.6%) occurred within the first year, decreasing from 38.4% in 2000–2003 to 26.7% in 2019–2022. Similarly, 242 deaths (47.4%) occurred within two years, showing a marked decline from 54.5% in 2000–2003 to 34.9% in 2019–2022.

Figure 1 illustrates the trend in 1-year and 2-year mortality. During 2000–2016, 1-year mortality showed a slight, non-significant increase (APC: 1.4, 95% CI: −0.2; 3.0), followed by a significant decrease in 2017–2022 (APC: −6.6, 95% CI: −13.1; −0.5). Similarly, 2-year mortality exhibited a non-significant decrease in 2000–2016 (APC: −0.3, 95% CI: −1.2; 1.7), followed by a significant decline in 2017–2022 (APC: −7.3, 95% CI: −12.3; −1.4).

Table 3 shows net survival at 1, 3, and 5 years, divided into three calendar periods (2000–2007, 2008–2016, 2017–2022) with follow-up through 31 December 2024.

One-year survival improved modestly from 63% (95% CI: 57–71) to 66% (95% CI: 56–75), while 3-year survival rose from 39% (95% CI: 32–46) to 42%, (95% CI: 28–51) and 5-year survival increased more substantially from 21% (95% CI: 14–28) to 30% (95% CI: 20–44) across the study periods.

Finally, an additional table reports data regarding drugs introduced into the province of Reggio Emilia starting from 2017.

## 4. Discussion

This population-based study describes temporal changes in the incidence of MBC over 23 years.

Among 10,966 invasive tumors, 511 (4.7%) were metastatic at diagnosis, with a decline from 6.4% in 2000–2003 to 3.8% in 2019–2022, despite COVID-19-related disruptions to screening. Although in 2020 we reported a rapid resumption of screening mammograms immediately after the 3-month lockdown [35], we cannot exclude a delay in new screenings in the following years.

Although higher screening participation is associated with a reduced proportion of metastatic diagnoses in our population, this interpretation should be made with caution. The literature has long highlighted that screening outcomes may be influenced by overdiagnosis and stage-shift bias, which can artificially increase early-stage incidence without necessarily translating into genuine reductions in advanced disease or mortality. Recent analyses have shown that overdiagnosis remains substantial across several cancer screening programs and is often underestimated due to methodological limitations in trial design and evidence synthesis [36]. Furthermore, contemporary modeling studies indicate that assumptions underlying stage-shift benefits—widely used to infer downstream reductions in metastatic disease—may be uncertain and prone to overestimating the real impact of early detection strategies [37]. Therefore, while our results are consistent with a potential benefit of sustained screening coverage, they should be interpreted in the context of this ongoing debate.

In high-income countries, MBC represents approximately 5–10% of new diagnoses, but rates vary geographically, reaching up to 30% in low- and middle-income settings [18,38]. While organized screening programs are highly effective in detecting early-stage disease [39,40], their impact on reducing metastatic presentations remains less clearly established [41,42]. Evidence suggests that mammography screening reduces the incidence of advanced-stage tumors, particularly where participation rates and screening quality are high. However, this benefit may not extend to all biologically aggressive or fast-growing breast cancers.

The positive effect appears most pronounced for tumors detectable within the screening intervals and in settings with high adherence, regularity, and consistent screening quality [43,44]. In our study, a marked decline in mortality was observed in 2017, when both 1-year (APC −6.6, 95% CI −13.1; −0.5) and 2-year mortality (APC −7.3, 95% CI −12.3; −1.4) decreased significantly. This turning point may partly reflect the introduction of CDK4/6 inhibitors for receptor-positive disease and pertuzumab for HER2-positive tumors, which became available in Italy that year and were rapidly adopted in Reggio Emilia. These therapies might have contributed to modifying the natural history of metastatic breast cancer, although causality cannot be established.

It is important to consider local healthcare delivery factors that may have contributed to the observed improvements. In Reggio Emilia, the diagnostic–therapeutic pathway is highly integrated, resulting in short time-to-treatment intervals and rapid uptake of newly approved targeted therapies. Multidisciplinary tumor boards, standardized clinical pathways, and consistent adherence to national guidelines further support timely and appropriate treatment choices. Additionally, the coordination between oncology services, primary care, and supportive care networks ensures continuity of management, which may enhance both early and long-term outcomes. These organizational characteristics likely amplified the benefits of screening and modern systemic therapies, contributing to the decline in early mortality observed.

Recent clinical advances have transformed the therapeutic landscape of breast cancer across molecular subtypes. Antibody–drug conjugates (ADCs) such as trastuzumab deruxtecan and sacituzumab govitecan have demonstrated remarkable efficacy, improving survival in both HER2-positive and HER2-low or triple-negative settings while maintaining acceptable safety profiles [45,46,47,48]. Other studies confirm the durable benefit of T-DM1 in residual disease, and trastuzumab deruxtecan is redefining HER2-targetability beyond traditional amplification thresholds [49]. In parallel, immunotherapy has become a first-line standard in metastatic triple-negative breast cancer: pembrolizumab-based regimens and atezolizumab combinations produce durable and clinically meaningful survival gains, particularly in PD-L1–positive tumors (Appendix A) [50,51,52].

In hormone receptor-positive/HER2-negative disease, CDK4/6 inhibitors have achieved substantial improvements in both progression-free and overall survival across both advanced and early-stage settings [53,54,55,56]. Collectively, these developments represent a shift toward precision-guided, biomarker-driven treatment integrating targeted agents, endocrine modulation, and immunotherapy. This evolution not only enhances survival outcomes but also redefines treatment sequencing and long-term disease control, underscoring the need for continuous refinement of therapeutic algorithms in breast cancer management. Additionally, advances in supportive care reflect the broader progress in integrated oncology management [57].

Survival outcomes have progressively improved. In the older period compared to the more recent one, the increase was 3% at 1 year, 3% at 3 years and 9% at 5 years from diagnosis. This improvement may be partly linked to the impact of new targeted therapies, including CDK4/6 inhibitors, next-generation anti-HER2 drugs and immunotherapy.

Population-based studies corroborate these findings: although overall survival and net survival are conceptually distinct metrics, population-based trends from the U.S. and UK provide a useful contextual benchmark. U.S. data show an age-adjusted improvement in 1-year overall survival from ~62% in 1988 to ~72% in 2015 [58], while UK data show 1-year and 5-year survival gains of ~2% at 1 year and ~5% at 5 years between 2000 and 2021 [59]. In our study, age-standardized 5-year net survival increased from 21% to 30% (the relative increase between the two periods corresponds to a ~43% rise from the first to the second period), indicating a trend of survival improvement broadly consistent with that observed in other high-income countries despite differences in survival metrics. Historically, five-year survival for advanced or biologically aggressive subtypes remained below 20%, with slower gains over time [60]. Thus, the large improvement observed here may reflect both earlier diagnosis and enhanced treatment efficacy. Overall, our findings align with the growing evidence that the progressive survival gains in breast cancer are primarily driven by advances in systemic therapy, complemented by optimized early management and supportive care. Organized screening has further improved outcomes by promoting earlier detection, while targeted treatments—including anti-HER2 agents, CDK4/6 inhibitors, and novel endocrine or immune therapies—have extended survival even in metastatic disease. These trends likely reflect a multifactorial synergy between improved diagnosis, therapeutic innovation, and comprehensive patient management [43,61,62,63,64].

### Strengths and Weaknesses

A limitation of this study is that metastatic breast cancer was defined to include both cases presenting with metastases at diagnosis and those developing metastases within six months after diagnosis. This choice reflects Cancer Registry conventions, whereby recurrences or newly detected metastases are classified only after this six-month interval. Another limitation of this study is that it is based on a single province in northern Italy. However, although the data refer to a single province, they can be generalized at least to the regions of Northern Italy. Indeed, although Italy has a public healthcare system that guarantees access to care for all citizens, there are differences in access, diagnosis, and treatment between the regions of Northern, Central, and Southern Italy. In Central and Southern Italy, participants in oncology research and oncology networks are fewer: the results show that overall survival rates are lower in Central and Southern Italy. The results of this study, referring to one region of Northern Italy, can be generalized to all regions of Northern Italy.

A second limitation was that for the most recent period (2017–2019), only three diagnosis years could be included to ensure a complete 5-year follow-up. This reduced case pool may limit statistical precision and introduce comparability bias in survival estimates across periods. Another limitation is represented by the lack of data on the molecular biology (receptor site and HER2 amplification), which may have had a different impact in women with metastatic breast cancer. This is an important gap that could be filled with a new study that also collects this information.

A key strength is that this is a population-based analysis, whereas most available data come from clinical trials. This design allows us to assess whether the benefits of screening and modern treatments are reflected in an unselected, real-world population, thereby strengthening the external validity of our findings.

A future study might consider reviewing the outcome data in metastatic cancers, adding information on molecular biology.

## 5. Conclusions

This population-based study from the province of Reggio Emilia provides compelling real-world evidence that, over more than two decades, the incidence of “early” metastatic breast cancer at diagnosis and early mortality both declined, accompanied by improved short-term survival. The pronounced reduction in 1- and 2-year mortality after 2017 coincides with the introduction of modern targeted therapies—most notably CDK4/6 inhibitors and pertuzumab—alongside the sustained benefits of organized mammography screening.

To contextualize the observed mortality decline, it is important to note that both CDK4/6 inhibitors and pertuzumab were rapidly incorporated into routine clinical practice in Reggio Emilia soon after their national approval. The oncology units of the province have long demonstrated a high level of adherence to national and international guidelines, supported by structured multidisciplinary tumor boards and standardized treatment pathways. As a result, eligible patients were promptly offered these therapies as first-line options in accordance with guideline recommendations. This timely and widespread adoption within an integrated care system reinforces the plausibility that the introduction of these agents contributed to the population-level improvements observed.

Over the past decade, the progressive introduction of targeted therapies—including modern anti-HER2 agents, CDK4/6 inhibitors, and, more recently, antibody–drug conjugates and immunotherapy—has improved the prognosis of metastatic breast cancer. While each class offers specific clinical advantages, their overall impact is best understood as a broader therapeutic shift toward more effective and personalized systemic treatments. This evolution, rather than the characteristics of individual drugs, aligns most closely with the temporal improvements in survival observed in our population.

Future research should aim to integrate molecular, therapeutic, and socioeconomic data to better characterize which patient groups derive the greatest survival benefit and to further optimize early detection and treatment strategies in routine practice.

## Figures and Tables

**Figure 1 cancers-18-00108-f001:**
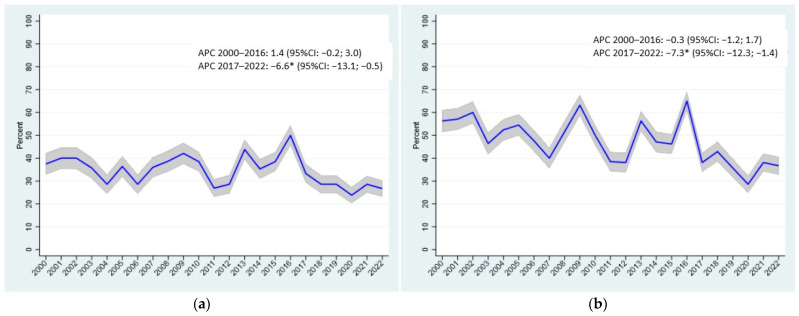
Province of Reggio Emilia. Number of deaths per year at 1 (**a**) and 2 years (**b**). *: *p*-value < 0.05.

**Table 1 cancers-18-00108-t001:** Province of Reggio Emilia. Number of breast cancers and “early“ metastatic cancers, by year.

	Cancers	Metastatic Cancers	
Year	Number of Cases	Number of Cases	%
2000	404	16	4.0
2001	415	35	8.4
2002	397	20	5.0
2003	421	28	6.7
2004	442	21	4.8
2005	439	22	5.0
2006	435	21	4.8
2007	449	25	5.6
2008	435	31	7.1
2009	431	19	4.4
2010	498	26	5.2
2011	500	26	5.2
2012	477	21	4.4
2013	494	16	3.2
2014	464	17	3.7
2015	521	26	5.0
2016	484	20	4.1
2017	522	21	4.0
2018	502	14	2.8
2019	522	14	2.7
2020	550	21	3.8
2021	578	21	3.6
2022	586	30	5.1

**Table 2 cancers-18-00108-t002:** Province of Reggio Emilia. Number of deaths at 1 and 2 years from “early“ metastatic cancers.

Year	Number of Metastatic Cancers	Deaths Within 1 Year of Diagnosis	% Deaths 1 Year/M1	Deaths Within 2 Years of Diagnosis	% Deaths 2 Years/M1
2000	16	6	37.5	9	56.3
2001	35	14	40.0	20	57.1
2002	20	8	40.0	12	60.0
2003	28	10	35.7	13	46.4
2004	21	6	28.6	11	52.4
2005	22	8	36.4	12	54.5
2006	21	6	28.6	10	47.6
2007	25	9	36.0	10	40.0
2008	31	12	38.7	16	51.6
2009	19	8	42.1	12	63.2
2010	26	10	38.5	13	50.0
2011	26	7	26.9	10	38.5
2012	21	6	28.6	8	38.1
2013	16	7	43.8	9	56.3
2014	17	6	35.3	8	47.1
2015	26	10	38.5	12	46.2
2016	20	10	50.0	13	65.0
2017	21	7	33.3	8	38.1
2018	14	4	28.6	6	42.9
2019	14	4	28.6	5	35.7
2020	21	5	23.8	6	28.6
2021	21	6	28.6	8	38.1
2022	30	8	26.7	11	36.7

**Table 3 cancers-18-00108-t003:** Net survival at 1, 3, and 5 years among “early“ metastatic breast cancer cases, by period.

Years	2000–2007		2008–2016		2017–2022	
	Survival (%)	95% CI	Survival (%)	95% CI	Survival (%)	95% CI
1	63	(57–71)	64	(57–70)	66	(56–75)
3	39	(32–46)	40	(32–47)	42	(28–51)
5 *	21	(14–28)	27	(21–34)	30	(20–44)

* Follow-up limited to cases diagnosed in 2017–2019.

## Data Availability

The data presented in this study are available on request from the corresponding author. The data are not publicly available due to ethical and privacy issues; requests for data must be approved by the Ethics Committee after the presentation of a study protocol.

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
