# Peer review of "Trends in Survival and Mortality of “Early" Metastatic Breast Cancer in Northern Italy Following the Introduction of Targeted Therapies"

_cancers, 2025, doi:10.3390/cancers18010108_

Round 1
Reviewer 1 Report
Comments and Suggestions for Authors
The respected authors summarized clinical outcomes of therapeutics regionally in their country.
In my opinion, they could improve this paper by adding a table that summarizes new drugs, such as nano-based drug delivery systems, alongside the others they have already mentioned. By using this strategy, your paper will become one of the pioneering studies on the effectiveness of this novel therapeutic strategy worldwide.
Author Response
The respected authors summarized clinical outcomes of therapeutics regionally in their country.
In my opinion, they could improve this paper by adding a table that summarizes new drugs, such as nano-based drug delivery systems, alongside the others they have already mentioned. By using this strategy, your paper will become one of the pioneering studies on the effectiveness of this novel therapeutic strategy worldwide.
RESPONSE: Thanks to the reviewer for the proposal. We've added a supplementary table summarizing the data usage and added an addition to the text.
Reviewer 2 Report
Comments and Suggestions for Authors
Introduction
The introduction is thorough and contains appropriate citations supporting global, national, and regional breast cancer trends. It effectively motivates the focus on metastatic disease and the influence of screening and modern therapies. Authors could consider briefly stating the knowledge gap this study addresses earlier in the introduction to alert the reader to the research rationale.
Methods
The population-based design and registry characteristics are clearly outlined. While the definitions of metastatic disease are appropriate, more details on metastasis ascertainment (e.g., imaging categories, pathology confirmations) would enhance clarity. The joinpoint regression approach is suitable, but it would help to add information on model selection criteria, permutation tests, and other joinpoint software settings. Authors should consider addressing whether tumor biology (i.e. ER/PR/HER2 status) was available and, if not, noting this explicitly as a limitation.
Results
While the depiction of mortality trends is useful, including confidence intervals or shading around trend lines may strengthen interpretability. Since therapies introduced in 2017 appear central to the interpretation, consider examining trends stratified by receptor status if data are available. Survival confidence intervals for the last period are wide and it would help to acknowledge limited follow-up in the narrative.
Discussion
The discussion provides a strong comparison to international trends and situates the findings within contemporary therapeutic landscapes. The authors thoroughly describe the potential impact of screening, targeted therapies, and immunotherapy as this enhances contextual depth. The section could be strengthened by expanding on specific local healthcare delivery factors (for example, time-to-treatment, treatment uptake) that may contribute to improved outcomes. The limitations paragraph is also brief, authors could consider expanding on data constraints such as lack of molecular subtype information, treatment data, and potential lead-time bias from screening. Overall, the conclusions are well-aligned with results and remain appropriately cautious regarding generalizability beyond similar high-income regions
Author Response
Comment 1 : The introduction is thorough and contains appropriate citations supporting global, national, and regional breast cancer trends. It effectively motivates the focus on metastatic disease and the influence of screening and modern therapies. Authors could consider briefly stating the knowledge gap this study addresses earlier in the introduction to alert the reader to the research rationale.
Response 1: Thanks to the reviewer for the suggestion. We've added a sentence to the text.
"Given the paucity of observational studies on the impact of new drugs in metastatic breast cancers, this study aims to fill a knowledge gap by attempting to provide some useful information for interpreting the efficacy of treatments".
Comment 2: The population-based design and registry characteristics are clearly outlined. While the definitions of metastatic disease are appropriate, more details on metastasis ascertainment (e.g., imaging categories, pathology confirmations) would enhance clarity.
Response 2: Thanks to the reviewer for the clarification. In fact, we hadn't specified how data on metastatic breast cancer is collected. We've added a sentence in the Materials and Methods section.
"Stage information is collected by registrars working at the CR. International registration rules state that registrars are not required to formulate diagnoses, but only to report the information they retrieve from medical records and letters to the doctor. This explains why our data always contain some UNKNOWN data, not because the patients were not staged, but because we were unable to retrieve it".
Comment 3: The joinpoint regression approach is suitable, but it would help to add information on model selection criteria, permutation tests, and other joinpoint software settings.
Response 3: Thanks for the suggestion. We added more information on jointpoint regression in the Materials and Methods section.
Comment 4: Authors should consider addressing whether tumor biology (i.e. ER/PR/HER2 status) was available and, if not, noting this explicitly as a limitation.
Response 4: We agree with the reviewer. A future study might consider reviewing the outcome data in metastatic cancers, adding information on molecular biology. We have added one sentence within the scope of the study.
Comment 5: While the depiction of mortality trends is useful, including confidence intervals or shading around trend lines may strengthen interpretability.
Response 5: Thanks for the comment. We added shading (I.C.) around trend lines in the Figure 1.
Comment 6: Since therapies introduced in 2017 appear central to the interpretation, consider examining trends stratified by receptor status if data are available. Survival confidence intervals for the last period are wide and it would help to acknowledge limited follow-up in the narrative.
Response 6: Thanks fo the comment. Unfortunately, data on receptor status are not available, but we could organize a future work also adding biological variables.
Comment 7: The discussion provides a strong comparison to international trends and situates the findings within contemporary therapeutic landscapes. The authors thoroughly describe the potential impact of screening, targeted therapies, and immunotherapy as this enhances contextual depth. The section could be strengthened by expanding on specific local healthcare delivery factors (for example, time-to-treatment, treatment uptake) that may contribute to improved outcomes.
Response 7: Thanks to the reviewer for the suggestion. We've added a section to the discussion about how the healthcare system is organized in our area.
"It is important to consider local healthcare delivery factors that may have contributed to the observed improvements. In Reggio Emilia, the diagnostic–therapeutic pathway is highly integrated, resulting in short time-to-treatment intervals and rapid uptake of newly approved targeted therapies. Multidisciplinary tumor boards, standardized clinical pathways, and consistent adherence to national guidelines further support timely and appropriate treatment choices. Additionally, the coordination between oncology services, primary care, and supportive care networks ensures continuity of management, which may enhance both early and long-term outcomes. These organizational characteristics likely amplified the benefits of screening and modern systemic therapies, contributing to the decline in early mortality observed."
Comment 8: The limitations paragraph is also brief, authors could consider expanding on data constraints such as lack of molecular subtype information, treatment data, and potential lead-time bias from screening. Overall, the conclusions are well-aligned with results and remain appropriately cautious regarding generalizability beyond similar high-income regions.
Response 8: Thank you for the suggestion. We've expanded the limitations section by adding the lack of molecular biology data and better described the generalizability of the results, which only applies to the northern regions of Italy, not to central and southern Italy. We've clarified this further in the discussion section.
Reviewer 3 Report
Comments and Suggestions for Authors
The manuscript presents a population-based analysis of metastatic breast cancer trends over a 23-year period in the province of Reggio Emilia, Italy, evaluating changes in incidence, mortality, and survival following the introduction of screening programs and modern targeted therapies. The topic is timely and highly relevant, providing real-world evidence on how precision oncology and organized mammography screening influence metastatic disease burden and patient outcomes. The study benefits from high-quality registry data and long-term follow-up; however, several methodological and interpretative aspects would benefit from clarification, particularly in defining metastatic status, exploring confounders, and contextualizing survival improvements relative to therapeutic adoption timelines. The detailed comments below are intended to help strengthen the clarity, rigor, and interpretability of the manuscript.
- Lines 36–38 – The definition of metastatic disease as “metastases within 6 months of diagnosis” needs justification, as this differs from the standard definition of de novo stage IV. This could misclassify early relapses.
- Lines 40–43 – APC estimates are presented without specifying the number of joinpoints detected. Clarifying the model output is essential for transparency.
- Lines 28–31 – The abstract compresses incidence, mortality, and survival into a single sentence. Consider separating these concepts to improve readability.
- Lines 44–48 – The conclusion implies causality between therapies/screening and improved outcomes; observational data cannot support causal inference and should be phrased more cautiously.
- Lines 123–124 – The method for determining metastatic status via medical record review needs detail (e.g., imaging criteria, pathology confirmation).
- Lines 126–131 – Justify allowing “up to four joinpoints,” as this choice affects trend interpretation over 23 years.
- Lines 136–139 – The decline from 6.4% to 3.8% metastatic diagnoses should be accompanied by statistical testing or confidence intervals to show significance.
- Line 138 – The unexpected increase to 5.1% in 2022 warrants discussion (COVID-19 delays? data completeness issues? random variation?).
- Lines 141–144 – Clarify whether mortality is all-cause or breast cancer–specific, which greatly impacts interpretation.
- Lines 154–158 – Specify the method used to calculate net survival (e.g., Pohar-Perme) and provide justification.
- Lines 156–157 – Survival differences are described as improvements, but no p-values or confidence intervals are provided to evaluate statistical significance.
- Line 160 – Limited follow-up for 2017–2019 cases may underestimate survival; discuss potential bias.
- Lines 164–170 – The conclusion that COVID-19 did not affect diagnosis needs stronger evidence or citation from regional screening programs.
- Lines 171–178 – The discussion links screening participation to metastatic reduction but should acknowledge the ongoing debate around overdiagnosis and stage-shift bias.
- Lines 182–187 – The temporal alignment between therapy introduction (2017) and improved mortality is suggestive but should not be overstated; note potential confounders.
- Lines 188–191 – Provide data on how widely CDK4/6 inhibitors and pertuzumab were adopted in the region to support claims of population-level impact.
- Lines 192–201 – The detailed listing of new therapies (ADCs, immunotherapies) is informative but too long; streamline to maintain focus on population-level trends.
- Lines 211–217 – The contrast between “modest” 1-year survival improvement and “substantial” 5-year improvement should be explained more precisely.
- Lines 218–221 – When comparing with U.S./UK survival trends, include standardized survival metrics or age-adjusted comparisons for accuracy.
- Lines 221–224 – The statement of a “~43% relative increase” in 5-year survival should include the calculation method to avoid misinterpretation.
- Lines 234–237 – While the authors claim generalizability to northern Italy, acknowledge differences in treatment access, referral patterns, and registries across regions.
- Lines 238–241 – Registry-based studies lack molecular subtype and treatment line data; explicitly stating this limitation will strengthen transparency.
- Lines 243–248 – Attributing survival improvements primarily to targeted therapies requires stronger evidence or acknowledgment of possible confounding from earlier diagnosis.
- Lines 252–254 – The conclusion states that precision oncology and screening have “reshaped the natural history” of disease; this should be softened given observational design.
- Lines 255–257 – Excellent recommendation for future work; however, specify which molecular or socioeconomic variables should be incorporated to enhance real-world modeling.
Author Response
Comment 1: Lines 36–38 – The definition of metastatic disease as “metastases within 6 months of diagnosis” needs justification, as this differs from the standard definition of de novo stage IV. This could misclassify early relapses.
Response 1: Thanks to the reviewer for the extremely important request for clarification. We have indicated metastatic breast cancer as required by the AIOM and ESMO guidelines: breast cancer is considered stage IV (de novo metastatic) if metastases are already present at the time of diagnosis, even if they appear within the first 6 months. The presence of metastases (beyond local lymph nodes) indicates systemic, not merely localized, disease, although the timing of onset can sometimes influence prognosis and classification. The key concept is that once the disease spreads to distant organs (lungs, bones, liver, brain), it is considered metastatic, regardless of the time of onset of metastases, but 6 months is often an important clinical indicator for prognostic classification. We clarified this concept in the materials and methods section.
Comment 2: Lines 40–43 – APC estimates are presented without specifying the number of joinpoints detected. Clarifying the model output is essential for transparency.
Response 2: Thanks for the suggestion. We added more information on jointpoint regression in the Materials and Methods section.
Comment 3: Lines 28–31 – The abstract compresses incidence, mortality, and survival into a single sentence. Consider separating these concepts to improve readability.
Response 3: Thank you for your suggestion. We have revised the abstract according to the reviewer's recommendations.
Comment 4: Lines 44–48 – The conclusion implies causality between therapies/screening and improved outcomes; observational data cannot support causal inference and should be phrased more cautiously.
Response 4: We completely agree with the Auditor. We expressed the concept more cautiously.
Comment 5: Lines 123–124 – The method for determining metastatic status via medical record review needs detail (e.g., imaging criteria, pathology confirmation).
Response 5: Thanks to the reviewer for the clarification. In fact, we hadn't specified how data on metastatic breast cancer is collected. We've added a sentence in the Materials and Methods section.
"Stage information is collected by registrars working at the CR. International registration rules state that registrars are not required to formulate diagnoses, but only to report the information they retrieve from medical records and letters to the doctor. This explains why our data always contain some UNKNOWN data, not because the patients were not staged, but because we were unable to retrieve it".
Comment 6: Lines 126–131 – Justify allowing “up to four joinpoints,” as this choice affects trend interpretation over 23 years.
Response 6 : Thanks for the comment. We added more information on jointpoint regression in the Materials and Methods section. The choice to allow the model up to 4 joint points is described in the chapter "Methodology for Characterizing Trends" from SEER (https://progressreport.cancer.gov/methodology). Specifically, for trends with 22-26 data points, they recommend setting the program with a maximum of 4 joint points.
Comment 7: Lines 136–139 – The decline from 6.4% to 3.8% metastatic diagnoses should be accompanied by statistical testing or confidence intervals to show significance.
Response 7: Thanks for the suggestion. We added confidence intervals.
Comment 8: Line 138 – The unexpected increase to 5.1% in 2022 warrants discussion (COVID-19 delays? data completeness issues? random variation?).
Response 8: Thank you for requesting clarification. Indeed, a post–COVID-19 effect cannot be excluded. We have refined this point in the manuscript using the following wording: “During the COVID-19 pandemic, all screening activities were temporarily suspended. In 2021 and 2022, screening progressively resumed, although some delays in new cancer diagnoses persisted. It will be important to assess whether the decline in metastatic tumors remains significant in 2023, thereby confirming the pre–COVID-19 trend.”
Comment 9: Lines 141–144 – Clarify whether mortality is all-cause or breast cancer–specific, which greatly impacts interpretation.
Response 9: We reported above that mortality is cause-specific. Since the Cancer Registry is closely linked to the Mortality Registry, it is easy for us to access the death records and verify the cause-specific mortality data.
Comment 10: Lines 154–158 – Specify the method used to calculate net survival (e.g., Pohar-Perme) and provide justification.
Response 10: Thanks for the comment. We added the method used to calculate net survival in the Materials and Methods section.
Comment 11: Lines 156–157 – Survival differences are described as improvements, but no p-values or confidence intervals are provided to evaluate statistical significance.
Response 11: Thanks for the suggestion. We added the confidence interval in the text.
Comment 12: Line 160 – Limited follow-up for 2017–2019 cases may underestimate survival; discuss potential bias.
Response 12: Thanks for the comment. We describe this limitation in the Materials and Methods section, and we discuss this potential bias in the Strengths and Weaknesses section.
Comment 13: Lines 164–170 – The conclusion that COVID-19 did not affect diagnosis needs stronger evidence or citation from regional screening programs.
Response 13: Thanks for the comment. We've reworded the sentence better.
Comment 14: Lines 171–178 – The discussion links screening participation to metastatic reduction but should acknowledge the ongoing debate around overdiagnosis and stage-shift bias.
Response 14: Thank you, we've added a sentence to the text that better reflects the concept suggested by the reviewer.
Comment 15: Lines 182–187 – The temporal alignment between therapy introduction (2017) and improved mortality is suggestive but should not be overstated; note potential confounders.
Response 15: We agree with the editor. We rewrote the sentence in a more hypothetical way.
Comment 16: Lines 188–191 – Provide data on how widely CDK4/6 inhibitors and pertuzumab were adopted in the region to support claims of population-level impact.
Response 16: Thanks, we've clarified this concept better in the text.
Comment 17: Lines 192–201 – The detailed listing of new therapies (ADCs, immunotherapies) is informative but too long; streamline to maintain focus on population-level trends.
Response 17: Thanks for the clarification. We've added a supplemetary table to simplify things.
Comment 18: Lines 211–217 – The contrast between “modest” 1-year survival improvement and “substantial” 5-year improvement should be explained more precisely.
Response 18: RE: Thanks for the suggestion: we've changed the text.
Comment 19: Lines 218–221 – When comparing with U.S./UK survival trends, include standardized survival metrics or age-adjusted comparisons for accuracy.
Response 19: Thank you for the comment. We have described this concept better in the Discussion section.
Comment 20: Lines 221–224 – The statement of a “~43% relative increase” in 5-year survival should include the calculation method to avoid misinterpretation.
Response 20: Thank you for the comment. The relative increase between the two periods was calculated as (30−21)/21, resulting in a 42.9% growth from the first to the second period. We have described it better in the text.
Comment 21: Lines 234–237 – While the authors claim generalizability to northern Italy, acknowledge differences in treatment access, referral patterns, and registries across regions.
Response 21: Thanks to the reviewer: we have clarified this part better in the text.
"Although Italy has a public health system that guarantees access to care for all citizens, there are differences in access, diagnosis, and treatment between the regions of northern, central, and southern Italy. In the central and southern areas, there are fewer participants in oncology research and fewer cancer networks: the results are that overall survival rates are lower in the central and southern regions of Italy. The results of this study, which referred to a region of northern Italy, can be generalized to all regions of northern Italy".
Comment 22: Lines 238–241 – Registry-based studies lack molecular subtype and treatment line data; explicitly stating this limitation will strengthen transparency.
Response 22: Thanks for the comment: this is a limitation of this study that we have better emphasized in the text.
"The study has some limitations: the first is related to the lack of data on the molecular biology (receptor site and HER2 amplification), which may have had a different impact in women with metastatic breast cancer. This is an important gap that could be filled with a new study that also collects this information".
Comment 23: Lines 243–248 – Attributing survival improvements primarily to targeted therapies requires stronger evidence or acknowledgment of possible confounding from earlier diagnosis.
Response 23: Thank you, we have clarified this part better in the text.
Comment 24: Lines 252–254 – The conclusion states that precision oncology and screening have “reshaped the natural history” of disease; this should be softened given observational design.
Response 24: We agree with the Auditor. We have rewritten this concept better.
Comment 25: Lines 255–257 – Excellent recommendation for future work; however, specify which molecular or socioeconomic variables should be incorporated to enhance real-world modeling.
Response 25: We agree with the reviewer. A future study might consider reviewing the outcome data in metastatic cancers, adding information on molecular biology and socioeconomic variables. We have added one sentence within the scope of the study.
Round 2
Reviewer 3 Report
Comments and Suggestions for Authors
The manuscript is now significantly improved, and I believe it meets the standards for publication. I have no further concerns and I recommend the paper for acceptance in its current form.